# Mediator-Related Symptoms and Anaphylaxis in Children with Mastocytosis

**DOI:** 10.3390/ijms22052684

**Published:** 2021-03-07

**Authors:** Knut Brockow, Katarzyna Plata-Nazar, Magdalena Lange, Bogusław Nedoszytko, Marek Niedoszytko, Peter Valent

**Affiliations:** 1Department of Dermatology and Allergology Biederstein, School of Medicine, Technical University of Munich, Biedersteiner Strasse 29, D-80802 Munich, Germany; 2Department of Paediatrics, Paediatric Gastroenterology, Allergology and Nutrition, Medical University of Gdańsk, 80803 Gdańsk, Poland; 3Department of Dermatology, Venereology and Allergology, Medical University of Gdańsk, 80210 Gdansk, Poland; m.lange@gumed.edu.pl (M.L.); bned@gumed.edu.pl (B.N.); 4Department of Allergology, Medical University of Gdańsk, 80210 Gdansk, Poland; marek.niedoszytko@gumed.edu.pl; 5Department of Internal Medicine I, Division of Hematology & Hemostaseology, Medical University of Vienna, 1090 Vienna, Austria; peter.valent@meduniwien.ac.at; 6Ludwig Boltzmann Institute for Hematology and Oncology, Medical University of Vienna, 1090 Vienna, Austria

**Keywords:** mast cells, mastocytosis, mediator-related symptoms, anaphylaxis, children

## Abstract

Mastocytosis is characterized by the pathological accumulation of mast cells (MC) in various organs. In these patients, MC may degranulate and thereby contribute to clinical symptoms, especially when a concomitant allergy is present. However, MC activation can not only be induced by high-affinity receptors for IgE, but also by anaphylatoxins, neuropeptides, IgG immune complexes, complement-components, drugs, products of bacteria or parasites, as well as physical factors such as heat, cold, vibration, stress, sun, or physical effort. Symptoms due to mediators released by activated MC may develop in adults suffering from systemic mastocytosis, but also evolve in children who usually have cutaneous mastocytosis (CM). Clinically, CM is otherwise characterized by typical brown, maculopapular skin lesions or mastocytoma associated with a positive Darier’s sign. Pruritus and flushing are common and blistering may also be recorded, especially in diffuse CM (DCM). Pediatric patients with mastocytosis may also have gastrointestinal, respiratory, and neurologic complaints. Although anaphylaxis is not a typical finding, pediatric patients with massive skin involvement and high tryptase levels have a relatively high risk to develop anaphylaxis. This paper reviews MC mediator-related symptoms and anaphylaxis in children with mastocytosis, with special emphasis on risk factors, triggers, and management.

## 1. Introduction

Mastocytosis is a heterogeneous group of disorders characterized by the pathological accumulation of mast cells (MCs) in one or more organs [1,2,3]. In a subset of patients, symptoms of MC activation are found [1,2,3]. Although mastocytosis can occur throughout life, there is a pronounced peak of incidence within the first two years of life [4]. Mastocytosis is rare, with a similar prevalence seen in different races. Overall, there is a slight predominance of boys in children and women among adults [5]. The disease may encompass only the skin (cutaneous mastocytosis, CM), or may involve internal organs such as bone marrow, liver, spleen, gastrointestinal tract, and lymph nodes (systemic mastocytosis, SM), with (SM+) or without (SM-) skin involvement [3,6,7,8]. In more advanced forms of disease, MC infiltration may lead to impairment or (sometimes even irreversible) damage of the affected organ systems [7,9,10,11]. CM is the most common form of mastocytosis in children and comprises over 90% of cases in this age group [8]. About 60–80% of pediatric patients may go into spontaneous remission in adolescence [2,5]. SM is infrequent in children and indicates a risk for persistence of the disease into adulthood [5]. Most children with SM have indolent SM, whereas advanced pediatric SM is very rare [1,5,12,13]. Complaints of patients with pediatric mastocytosis, apart from cosmetic consequences of the skin lesions, are mostly on symptoms caused by mediators released from MCs. These symptoms may be mild, moderate, or rarely life-threatening. MC mediator-related events may present as sudden attacks, intermittently, or may be observed repeatedly during the course of disease. These symptoms can be recorded in children with CM and SM [5,14,15,16].

## 2. Mediators Released from MCs

MCs were described by Paul Ehrlich, Noble Prize winner and one of the grandmasters of modern immunology, who was born in 1854 in Strzelin near Wroclaw (now in Poland, Prussia in 1854) [17]. MCs contain granules packed with vasoactive and immunomodulatory mediators that can be visualized by metachromatic staining [6,18]. Different pathways may lead to MC activation and degranulation. One important signaling pathway is activated by high-affinity receptors for IgE. This activation pathway is particularly relevant to immediate type allergic reactions [19]. However, multiple other triggers of MC activation have been described, including certain cytokines, anaphylatoxins, neuropeptides, IgG immune complexes, complement, drugs, radiocontrast media, products of bacteria, or parasites [2]. Physical factors such as heat, cold, vibration, stress, sun, physical effort may also contribute to or induce MC activation and act as co-factors in allergic and anaphylactic reactions [20]. MCs contain two types of mediators—the first stored in the cytoplasmic granules (responsible for the early phase of allergic reaction) and the second synthesized during and after MC activation (responsible for the late phase reaction) [21]. Preformed mediators include histamine, tryptase, chymase, carboxypeptidase, and heparin. Mediators synthesized during MC activation are lipids, such as prostaglandin D2, leukotriene C4, platelet-activating factor, and cytokines, including interleukin-5 (IL-5), IL-6, transforming growth factor-β, or tumor necrosis factor α [21]. However, it should be mentioned that most of these cytokines have been detected in murine MCs rather than in human MCs. In human MCs, other MC-derived cytokines and chemokines may play a role in MC-related functions and particularly in the mastocytosis context. These compounds include, among others, vascular endothelial growth factor (VEGF), IL-8, monocyte chemoattractant protein-1 (MCP-1), and oncostatin-M [22,23,24,25].

Mediators released from MCs evoke vascular, gastrointestinal, bronchial smooth muscle, and endothelial cell responses leading to vasodilatation and microvascular permeability, stimulation of mucus glands, and afferent nerve endings [26]. Histamine is one of the most important mediators released by degranulating MCs [27,28]. Histamine can cause symptoms in the skin (pruritus, urticaria, reddening/flushing, angioedema), in the respiratory system (runny nose, sneezing, coughing, wheezing, bronchospasm), in the gastrointestinal system (abdominal cramps, diarrhea, vomiting, gastric hypersecretion), and the circulatory system (tachycardia, hypotension, anaphylactic shock) [29]. Tryptase is the most common, sensitive, and relatively specific mediator of MC activation and has found application in laboratory diagnostics. Heparin-a polysaccharide, inhibits the blood clotting at a stage when prothrombin is converted into thrombin and thus prevents the formation of blood clots in the blood vessels. It is an anticoagulant and intensifies bleeding tendency. Prostaglandin D_2_ (PGD_2_) is the major eicosanoid product of MCs and is released in large quantities during anaphylaxis and asthma attacks. Patients suffering from mastocytosis produce excessive amounts of PGD_2_, which causes flushing, vasodilation, hypotension, and syncopal episodes. PGD_2_ can also induce increased motility of the gastrointestinal tract, manifested by diarrhea [29]. Leukotriene C4 (LTC4) plays a role in inflammation, immunological functions, and maintaining biological homeostasis. LTC4 induces mucus secretion in airways and long-lasting hypotension, reduces myocardial contractility and coronary blood flow, and interferes with the motility function of the gastrointestinal tract (abdominal pain, diarrhea) [29]. Platelet-activating factor (PAF) causes bronchoconstriction and increases the permeability of blood vessels. MC-derived cytokines may also attract other inflammatory cells from the circulation, such as neutrophils, eosinophils, and T lymphocytes, to extend the local inflammatory response [30,31,32,33]. Nevertheless, the assessment of the exact role of all these mediators in the pathogenesis of mastocytosis requires further studies.

## 3. Mediator-Related Symptoms in Children with Mastocytosis

### 3.1. Cutaneous Symptoms

In most children, mastocytosis infiltrates are limited to the skin and the diagnosis is thus CM. In these patients, CM often presents as maculopapular brown skin lesions (MPCM), but may also present as erythroderma and pachyderma typical for diffuse cutaneous mastocytosis (DCM) or up to four nodular or plague lesions termed mastocytoma [8]. Regardless of the subform, CM may be associated with MC mediator-related symptoms in the skin such as pruritus, flushing, and blistering [34,35,36]. Generally, the probability for having these symptoms is more likely in patients with indurated skin lesions containing packed accumulations of MCs typically seen in DCM. The Darier’s signs is pathognomonic for mastocytosis patients with skin lesions, including CM, and presents as erythema and edema, which is observed within a few minutes after mechanical irritation of lesional skin [8]. The Darier’s sign is present in up to 90% of all patients with mastocytosis [5]. However, absence of the Darier’s sign does not exclude the diagnosis of mastocytosis [1,15]. The Darier’s sign should be elicited gently in case of large mastocytoma, as it may result in flushing and hypotension [8,37]. Pruritus is a very common skin symptom and occurs in at least half of the children with mastocytosis (46–79%) [5,14,15,36]. Flushing is also relatively common in childhood CM (20–65%) [1,5,36]. In some of the patients, flushing precedes the development of anaphylaxis [16,38]. Blisters occur preferentially in children under 2–3 years of age, often in the context of DCM. Overall, blistering is reported in about 25–35% of children with mastocytosis [5,15,36,39].

### 3.2. Extracutaneous Systemic Symptoms

Although MC mediator-related symptoms in pediatric mastocytosis are typically cutaneous, children may also suffer from gastrointestinal (up to 40% of children), respiratory (13%), and neurologic (6–18%) symptoms [5,15,36,39]. Gastrointestinal symptoms comprise abdominal pain, cramps, bloating, diarrhea, nausea, and vomiting [1,5,38,39]. Histamine (and other mediators) increases gastric acid secretion, leads to hyperacidity, and may induce gastroduodenal ulcer disease. Patients with moderate or severe symptoms should be referred to a gastroenterologist and further tests, including endoscopy, should be considered. Respiratory symptoms caused by MC mediators are not commonly seen in children with mastocytosis (<13%) [15]. These symptoms may include nasal pruritus, rhinorrhea, bronchoconstriction, wheezing, stridor, and cough [1,28]. Neurologic symptoms reported in patients with pediatric mastocytosis include, among others, aggressive behavior, anxiety, depression, and loss of concentration [34,40]. Autistic behavior has also been described in children with mastocytosis [41]. Cardiovascular symptoms, including tachycardia, hypotension, shock, or collapse are very rare [14,36]. These symptoms occur mainly in children with extensive skin lesions and elevated serum tryptase level [14,15,42]. Similarly, musculoskeletal symptoms such as pain, osteopenia, osteoporosis, and pathologic fractures are rarely found in children with mastocytosis (6–13%) [5,39]. Constitutional symptoms such as fever, fatigue, or weight loss are also rare in children. All these symptoms occur in SM rather than in patients with CM [1,4,37].

### 3.3. Anaphylaxis

In children with mastocytosis, the reported percentage of anaphylaxis ranges between 0% and 9% (Table 1) [43,44]. Thus, the prevalence of anaphylaxis described in childhood mastocytosis is higher than the frequency reported in the general pediatric population (0.02–0.05%) [45]. On the other hand, anaphylaxis in childhood patients with mastocytosis is less frequently recorded compared to anaphylaxis in adults with mastocytosis [43,46].

### 3.4. Triggering Factors of Mediator-Related Symptoms and Anaphylaxis

Systemic symptoms induced by MC mediators can occur spontaneously or are provoked by certain stimuli, such as hymenoptera insect stings, aspirin allergen exposure, contrast media, surgery or endoscopy, and infections (bacteria, viruses, others).

The most prominent trigger of anaphylaxis in adults with mastocytosis is insect venom [43,48,49], whereas surprisingly in children with mastocytosis, hymenoptera stings are not a frequent trigger and there is no evidence for an increased risk in this patient group (Table 1) [34,50]. Correspondingly, in the pediatric populations with insect sting anaphylaxis presenting in the allergy clinic for allergy workup and in unselected patients presenting for acute treatment of anaphylaxis in an emergency department, the diagnosis of mastocytosis is very rare [50,51]. In one study of children with MPCM, hymenoptera stings occurred in 51% of 43 children, with no reports of anaphylaxis [44].

Foods and drugs have also been reported by patients as potential triggers of anaphylaxis in adults as well as in children with mastocytosis [34,50,52,53,54]. It has been described that children with mastocytosis may have a higher risk to develop adverse reactions to vaccines than healthy controls [55,56]. However, with a few exceptions, no anaphylaxis was reported after vaccination in childhood patients with mastocytosis [43,55,56]. Anesthesia may rarely be associated with MC mediator release [54]. One study reported perioperative anaphylaxis in one out of 50 anesthetic procedures in 48 children with mastocytosis during general anesthesia for major surgery [47]. The child who had developed an anaphylactic reaction, was able to tolerate the same anesthetic procedure at a later time. In another study, no serious adverse events occurred in children with mastocytosis undergoing surgical procedures [57].

Physical stimuli, such as jumping into cold water has been reported anecdotally as a trigger for anaphylaxis [36,43]. If no potential trigger is found, idiopathic anaphylaxis is diagnosed. Idiopathic anaphylaxis has also been reported in a minority of adults with anaphylaxis [58], but is the primary etiology reported in pediatric mastocytosis [14,34]. It has to be considered, however, that idiopathic anaphylaxis is more a provisional than a final diagnosis regarding the lack of identified trigger factors and before it is diagnosed, all potential known triggers which can be tested should be ruled out. It may well be that trigger factors do exist, but have not been tested for or yet described.

### 3.5. Risk Factors for Anaphylaxis in Children with Mastocytosis

Anaphylaxis can also occur in a child with mastocytosis and is difficult to predict reliably [36,59]. However, important risk factors have been identified for anaphylaxis in children (Table 2) [60,61]. In one study, the extent and density of skin lesions and higher serum tryptase values have been identified as risk factors for anaphylaxis in children with mastocytosis [43]. This is in agreement with another study, where the maximum number of maculopapular cutaneous mastocytosis lesions and the skin symptoms, such as itching, blistering, and flushing were statistically significant predictors of occurrence of systemic symptoms in a multivariate linear regression analysis [15]. Furthermore, DCM, the most extensive form of cutaneous mastocytosis, which is associated with high serum tryptase values, has been reported as a risk factor for more severe anaphylaxis [14]. A study with 111 children suffering from mastocytosis with skin involvement (mostly CM) looked for predictors for severe MC activation episodes, of which most appear to fulfil the definition of anaphylaxis [14]. In this study all 12 children with severe symptoms who required hospitalization had extensive cutaneous disease, more than 90% of the body surface area involved, and significantly higher levels of serum tryptase compared with the other children without such severe skin involvement and MC burden. Nine out of nine children (100%) with DCM had severe MC mediator-induced symptoms requiring hospitalization. Only three out of 102 children with nodular or maculopapular cutaneous mastocytosis required hospitalization because of the severity of mediator-related symptoms. Another analysis of this study revealed that blistering episodes are a risk factor for acute severe reactions requiring hospitalization [16]. Thus, children with severe skin involvement in CM, high serum tryptase levels, and DCM are at higher risk to develop anaphylaxis, particularly during blistering episodes, whereas the risk in the other patient groups seems to be low and might not be substantially increased as compared to the general population (Table 2) [14,15,34,43,46]. This is also substantiated by a higher prevalence of anaphylaxis in those studies with children more affected by skin involvement and serum tryptase levels as compared to those with uncomplicated limited maculopapular cutaneous mastocytosis (urticaria pigmentosa) [14,16,45].

### 3.6. Hereditary Alpha-Tryptasemia as Risk Factor for Mediator-Related Symptoms and Anaphylaxis

Recently, hereditary alpha-tryptasemia (HαT) has been described, a genetic trait with duplicated TPSAB1 germline copy numbers leading to elevated basal serum tryptase [62,63,64], but also to MC mediator-related symptoms. It can be found in about 5% of unselected healthy individuals [63,64,65,66,67]. The prevalence of certain symptoms related to MC activation has been described to be higher in HαT carriers than in the general population, with urticaria/angioedema in 51%, skin flushing/pruritus in 32–55%, and irritable bowel syndrome or food intolerance in 28–49% of patients [59,64]. The presence of HαT was identified as a heritable genetic risk factor for severe grade IV hymenoptera venom anaphylaxis, for idiopathic anaphylaxis and for SM in comparison to controls [67]. The prevalence of HαT was similar in patients with hymenoptera venom allergy with and without mastocytosis, however their clinical reactions were more severe. In a study assessing TPSAB1 germline copy number variants in 180 mastocytosis patients, 180 sex-matched control subjects, 720 patients with other myeloid neoplasms, and 61 additional mastocytosis patients of an independent validation cohort, α-tryptase encoding TPSAB1 copy number gains corresponding to HαT were identified in 17.2% of mastocytosis patients as compared to only 4.4% of the control population (*p* < 0.001) [65]. This pronounced difference indicates a possible pathogenic role of TPSAB1 copy number gains in the evolution of mastocytosis. Patients with HαT had highly significantly increased tryptase levels as compared to patients without HαT, which was independent of the mast cell burden. In patients with mastocytosis, hymenoptera venom hypersensitivity reactions and severe anaphylaxis were much more frequent in patients with HαT as compared to HαT-negative mastocytosis patients. The study concluded that HαT is an important biomarker for a predisposition to develop severe anaphylaxis in mastocytosis (Figure 1)

In fact, we speculate that an increased amount of tryptase may also lead to an increased load of other mediators (per mast cell) that are rapidly released by these cells through an anaphylactic event. An alternative explanation, recently been proposed by Le et al. [66], is that hetero-tetramers of tryptase, composed of 2α- and 2β-tryptase protomers (α/β-tryptase), activate protease-activated receptor-2 on smooth muscle, neurons, and endothelium and increase mast cell susceptibility to vibration-triggered degranulation by cleaving the α subunit of the EGF-like module-containing EMR2. The severity of the anaphylaxis in HAT may be caused by the increased vascular permeability which may lead to rapid hypotension caused by systemic allergic reaction. The study by Lyons et al. showed that tryptase hetero-tetramers selectively cleave PAR-2 (protease-activates receptor 2) on human vascular endothelial cell monolayers which increases vascular permeability [67].

Sabato et al. diagnosed HαT in three of four family members suffering from recurrent episodes of abdominal cramping and diarrhea who were diagnosed as MC activation syndrome (MCAS) [68]. These studies so far included adult patients, whereas little is known about symptoms and the severity of symptoms in childhood carriers of HαT in CM. In the pediatric population, gastrointestinal symptoms, urticaria, and hymenoptera venom allergy are rare. Mean serum tryptase, measured in 131 children aged 3 months to 18 years and hospitalized due to temporary self-resolving symptoms, was low (2.8 ± 2.2 ng/mL) with an upper reference limit of 7.2 ng/mL [69]. Further studies are required to indicate, if analysis of TPSAB1 copy numbers may become a part of the standard examination performed in children with anaphylaxis and MC activation syndrome. The diagnosis of HαT should be considered when basal tryptase level is ≥8–10 ng/mL [63].

## 4. Diagnostic Testing

A step-wise approach in children with suspected mastocytosis is recommended [2,70]. A thorough medical history collected from the patient includes the date of the first symptoms, duration of the disease, manifestations, localization, number of skin lesions (increase/decrease), their morphology, variability, factors provoking mediator release, related acute clinical symptoms, accompanying symptoms, and family history (mastocytosis in the family). A physical examination of the child should assess the physical development (weight, length), should be carried out with special attention to the inspection of the entire skin surface, palpation of all groups of lymph nodes (unexplained lymphadenopathy?), and abdomen examination (hepatosplenomegaly?) [40]. In children with skin lesions, histopathological examination of the skin may be indicated to establish the final diagnosis [8].

In all children, baseline serum tryptase levels should be obtained because a very high serum tryptase level (over 100 ng/mL) may be indicative of a high MC burden and/or the presence of SM [7,70,71,72,73]. Serum tryptase levels are usually within a normal range or slightly elevated in CM whereas tryptase levels are clearly elevated in most patients with SM and DCM [10,36,42,74,75]. Interestingly, patients with severe MC mediator-related symptoms or MCAS do not always have an elevated baseline serum tryptase level. Moreover, worsening of symptoms does not always correlate with an increase of tryptase level [4,35]. In children with mastocytosis, baseline level should be determined at least 24–48 h after the complete resolution of severe mediator-related symptoms [7,70]. In those children with elevated tryptase levels above 10 µg/l and severe MC-mediated symptoms, it is prudent to determine extra copies of the alpha tryptase gene (TPSAB1), if the test is available, to better understand the genetic basis of these symptoms.

Other recommended laboratory tests include a complete blood count with white blood cells, differential counts, and liver function tests [7,40]; however, abnormalities in blood counts and serum chemistry parameters are uncommon in childhood CM [34]. If hepatosplenomegaly is present, children should have an abdominal ultrasound or computer tomography. Organomegaly is a strong indicator of systemic disease [9]. Therefore, in children with mastocytosis associated with organomegaly, after excluding other causes, a detailed hematology work-up and the determination of *KIT* D816V mutation in peripheral blood (PB) has been recommended [13,70,72,73]. Bone marrow examination including histopathology, cytology, and flow cytometry is considered in children with *KIT* D816V mutation in peripheral blood (PB) and/or with suspected advanced SM [4,12,13,75].

Children with mastocytosis should be followed-up regularly, also when they reach adulthood.

## 5. Treatment

Children with mastocytosis may require a multidisciplinary approach and should best be diagnosed and followed-up by centers with experience in managing the disease. As causal drug therapy is not available, mainly symptomatic treatment is recommended. Education of patients and their family members form the basis of mastocytosis management. Patients and parents should be alerted about possible triggering factors (e.g., physical factors, alcohol, infections, certain drugs, stress) and risk situations (e.g., hymenoptera sting, general anesthesia) that may potentially induce mediator-related symptoms. Nevertheless, avoidance of all agents in a long list of all potential triggers reported in the literature is not an appropriate approach. Rather, the patients and parents have to learn to avoid classical triggers and to strictly avoid all additional triggers against which the childhood patient did react in the past.

### 5.1. Topical Therapy of Skin Symptoms

Local skin care with emollients may be applied in children with dry skin to prevent itching and scratching. Emollients decrease trans-epidermal water loss, moisture the skin, and seal the epidermal barrier which leads to the reduction of pruritus and decreases the sensitivity to physical stimuli [76]. Disodium cromoglycate at a concentration of 0.21% to 4% cream may be used to reduce pruritus [77,78]. Topical steroids against blistering should be used cautiously in children, and should only be applied in short-term therapy and for limited skin areas, because of numerous side effects of this therapy, particularly the risk of skin atrophy and adrenal suppression [79]. In infants, mild or medium potency corticosteroids are usually effective. Therapy with mometasone furoate 0.1% cream applied once daily to areas prone to the development of blistering in a neonate with DCM resulted in essential improvement [79]. The highly potent corticosteroid clobetasol propionate under occlusion has been used for solitary mastocytoma associated with recurrent blistering and pruritus, but may lead to skin atrophy [80]. Asymptomatic mastocytomas regress spontaneously and for this reason, local therapy is not necessary [81]. Mastocytoma associated with flushing or hypotension after mechanical irritation may be treated by surgical excision [80]. Therapy with topical antibiotics such as mupirocine or fucidic acid is recommended in children with denuded skin areas to prevent skin infections, particularly in those with blistering and DCM [34]. Another therapeutic option is pimecrolimus, a calcineurin inhibitor that has shown effects in children with symptomatic CM [82,83]. In particular, effective and safe therapy of MPCM and mastocytoma with 1% pimecrolimus cream applied twice daily on skin lesions was reported [82].

It has also been shown that narrow-band UVB (NB-UVB), as well as PUVA, reduces pruritus, and relieves skin symptoms in adults with both CM and ISM [84]. Also, in very selected children with extensive skin lesions and/or severe MC mediator-related symptoms unresponsive to antimediator therapy, phototherapy may be considered [4,77,85]. However, generally phototherapy and photochemotherapy PUVA (UVA in combination with psoralen) are not recommended in pediatric patients with mastocytosis. First, the tendency toward spontaneous regression of skin lesions around puberty is observed in a majority of children. Second, the overall period of symptom reduction is relatively short. Furthermore, little is known about the long-term safety of phototherapy in children [86,87]. PUVA therapy is associated with the risk of skin cancers (squamous skin carcinoma and melanoma), cataracts, and hepatotoxicity (psoralen-induced). Therefore, a very critical risk-benefit analysis should precede the decision of whether to apply any form of phototherapy in a child with CM.

### 5.2. Systemic Therapy of MC Mediator-Induced Symptoms

Oral antihistamines are the mainstay of anti-mediator therapy in pediatric mastocytosis. In symptomatic patients, first-line treatment is based on second-generation antihistamines, which block H_1_ receptors (cetirizine, desloratadine, fexofenadine, levocetirizine, loratadine and rupatadine, bilastine). They are applied to reduce itching and flushing, as well as general symptoms occurring as the result of the release of MC mediators. Age-appropriate doses are usually recommended, but sometimes it is necessary to administer up to four times the basic dose for age. First-generation antihistamines (chlorpheniramine, diphenhydramine, hydroxyzine, ketotifen, azelastine) are less commonly used due to their sedative effect; however, they are more effective in children with severe pruritus [2,37,86,88,89,90]. Antihistamines that block the H_2_ receptor are often used to alleviate gastrointestinal ailments, but currently for gastrointestinal symptoms, proton pump inhibitors and/or oral cromolyn sodium are applied with equal frequency [2,37,88,91].

In case of blistering or systemic symptoms unresponsive to H_1_-antihistamines, cromolyn sodium, and proton pump inhibitors, leukotriene antagonists may be offered. Oral steroids are very effective, but should only be taken for a short term [34]. Long-term use may cause numerous side effects, some of which (like skin changes, hypertension, obesity, osteoporosis, osteonecrosis) may be irreversible [86].

In the rare pediatric cases with osteopenia and osteoporosis vitamin D3 preparations, calcium preparations, and bisphosphonates are recommended [2,3].

### 5.3. Management of Anaphylaxis

In children with mastocytosis, acute management depends on the severity of the reaction. The first-line medication is adrenaline (epinephrine), administered at a dose of 0.01 mg/kg in children. In Germany a prefilled autoinjectors are available with a dose of 0.15 mg for children from 7.5–15 kg up to 25–30 kg, depending on the license of the autoinjector [92]. Adrenaline should be administered intramuscularly and, if necessary, can be repeated every 5 to 15 min [61]. Supporting measures for cardiovascular or respiratory reactions are high-flow oxygen, positioning the patient (e.g., Trendelenburg position with the lower extremities elevated for hypotension), volume replacement as well as inhaled adrenaline or beta-agonists (e.g., salbutamol). For cutaneous or gastrointestinal reactions, H_1_ and H_2_ antihistamines +/- steroids and monitoring are normally sufficient. After stabilization, blood chemistry parameters, blood counts, and a serum tryptase level should be determined in all patients. A massive event-related increase in tryptase may lead to the diagnosis of an MCAS.

Emergency preparedness is important in those children and in those situations with higher risk for anaphylaxis. Thus, mastocytosis children with a huge burden of MC in the skin (>45% of skin surface and/or a maximum density of lesional skin in relation to non-involved skin of >15%), those with high serum tryptase levels (>20 ng/mL), and cases with diffuse CM are at high risk for anaphylaxis (Table 2). Acute blistering episodes indicate a temporarily increased releasability of MCs and should be monitored carefully, and in the case of complications, the patient should be hospitalized. High-risk patients and family members should receive a prescription of a self-injectable adrenaline autoinjector and should carry it at all times. In children with less extensive mastocytosis and without a previous episode of anaphylaxis or individual risk factors (e.g., systemic peanut allergy) this is not required, because for those children the risk of anaphylaxis is low. Emergency kits for self-application may also contain an H_1_ antihistamine (e.g., dimethindene or cetirizine) and a corticosteroid (e.g., betamethasone) [92]. It is not sufficient to prescribe an emergency kit without appropriate instruction on how to recognize anaphylaxis and on how to use the emergency medication, best with an adrenaline injector trainer available from the producers, or even better after a structured anaphylaxis educational program [93]. A mastocytosis passport listing information on acute treatment, prescribed emergency medication, and drugs to avoid or to be given under medical supervision is very helpful for the patient.

Vaccination and anesthesia can be performed in children with mastocytosis. In those patients, precautions may be taken and emergency preparedness should be ensured [47]. The occurrence of reactions to vaccines in childhood mastocytosis has been predominantly reported after a hexavalent vaccine administration and injection of single vaccines has been suggested in children with diffuse cutaneous mastocytosis [55,56]. If this is necessary and appears unclear considering the mild severity of the reactions reported so far, however, administration of the first dose of vaccine to children with mastocytosis in a controlled clinical setting with monitoring of the patients for a prolonged observation interval (e.g., 1–2 h after injection or overnight in case moderate or severe reactions occur) is advisable. Although there is some disagreement on whether to recommend premedication with prophylactic anti-mediator-type therapy in every patient with mastocytosis before anesthesia it is advisable to recommend such prophylaxis at least in those with an unknown or a high risk to develop anaphylaxis [54]. Most pre-medications include an H_1-_ and H_2_-antihistamine and corticosteroids. Although premedication may not be needed in all children with mastocytosis, we recommend to consider in all cases and to administer in all with previously documented anaphylaxis. It is cautious to consider choosing those anesthetic drugs with low capacity to elicit MC degranulation as well as to use drugs with known tolerance by the individual child. Thus, a meticulous preparation and understanding of anesthetic implications are advised. It may also be important to reduce stress and anxiety, sometimes by the use of benzodiazepines, and to avoid temperature changes and mechanical stimulation for the patient [57]. Finally, it is important to inform the vaccination team and anesthesia team about the presence of a mast cell disease and, if known, the presence of a concomitant allergy.

Recently, omalizumab has been proposed for mastocytosis patients to treat severe mediator-related symptoms and to prevent anaphylaxis [3,94,95,96,97,98,99,100,101,102,103,104]. The drug was indeed found to be effective in the prevention of recurrent episodes of idiopathic anaphylaxis in adults with mastocytosis [94,95,96,97,98,99,100,101,102,103,104], but has to be licensed for use only in children over 12 years of age. The results of the XOLMA study has shown that omalizumab was safe and improved mastocytosis symptoms like diarrhea, dizziness, flush, and anaphylactic reactions [103]. First long-term follow-ups are available with improvement of clinical symptoms in two patients with SM and recurrent anaphylaxis over an observation period of 12 years [102]. Several case reports have suggested the usefulness of omalizumab in SM [95] and CM in adults [96], but also in adolescents [97], and children [98]. Omalizumab efficacy is not only restricted to allergic responses, but also abrogates non-allergic chronic spontaneous urticaria. It is believed that this effect is related to the downregulation of FcεRI receptors on basophils and mast cells, however, its detailed mechanism of action is not currently understood [105]. On the other hand, most anaphylactic reactions in patients with mastocytosis and/or MCAS can now be explained by (often occult forms of an) IgE-dependent allergy. Although the evidence is based only on uncontrolled observations in a small number of patients, this drug is likely to be effective in the prevention of recurrent anaphylaxis in children with or without mastocytosis [100]. Indeed, omalizumab has been described to prevent recurrent anaphylaxis attacks in a child with increased basal serum tryptase levels without mastocytosis [101]. The first approach, however, may be to administer high doses of anti-mediator type therapy (H1- and H2-antihistamines, cromoglycate, montelukast) and to study efficacy and tolerability of these drugs before offering omalizumab, which usually works in most patients (authors’ personal experience).

## Figures and Tables

**Figure 1 ijms-22-02684-f001:**
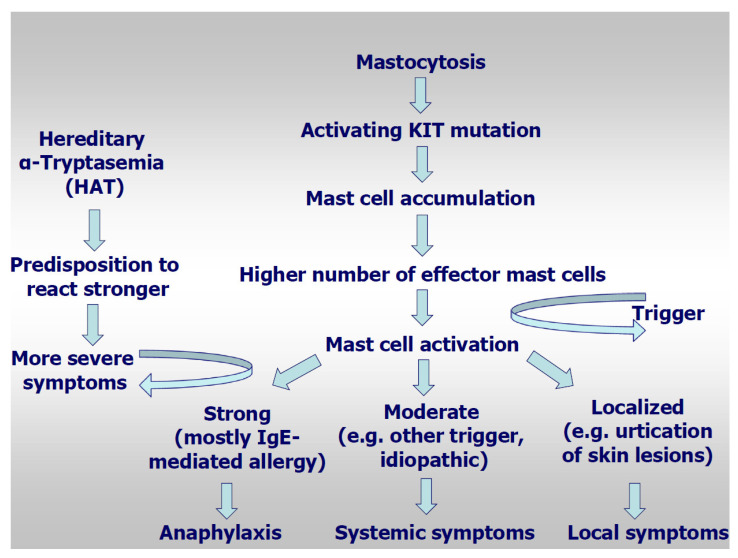
Model for pathogenesis of anaphylaxis in mastocytosis.

**Table 1 ijms-22-02684-t001:** Frequency of anaphylaxis in children with mastocytosis in the literature.

Study *	Year	No	Frequency	Elicitors
Gonzales [46]	2007	47	3 (6.4%)	Food, idiopathic
Brockow [43]	2008	46	4 (9%)	Food, vaccination, cold water, idiopathic
Alvarez-Twose [14]	2012	111	4 (3.6%)	Heat, skin rubbing, fever
Lange [36]	2013	101	7 (7%)	Clindamycin, ketamine, gadolinium contrast, stress, idiopathic
Barnes [15]	2014	67	1 (1,5%)	NA
Matito [47]	2015	48	1 (2.0%)	Perioperative
Lange [39]	2017	102	2 (2%)	Ketamine, idiopathic
Heinze [44]	2017	43	0 (0%)	NA
**Studies combined**		**565**	**22 (3.9%)**	

* First authors; NA = not available.

**Table 2 ijms-22-02684-t002:** Risk factors for anaphylaxis in children with mastocytosis (adapted from [60]).

Criterium	Risk Factor Outcome	Study
Extent and density of skin lesions	Anaphylaxis in those with extent >45% and density >15%	Brockow et al. [43]
Serum tryptase	Significantly elevated in patients with anaphylaxis	Brockow et al. [43]
	Correlation with severity	Alvarez-Twose et al. [14]
Skin involvement	>90%: risk factor for hospitalization *	Alvarez-Twose et al. [14]
Blistering	Risk factor for hospitalization *	Brockow et al. [16]
Diffuse cutaneous mastocytosis	Risk factor for hospitalization *	Alvarez-Twose et al. [14]

* Hospitalization because of severe mast cell activation episodes.

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
