# Peer review of "Mediator-Related Symptoms and Anaphylaxis in Children with Mastocytosis"

_ijms, 2021, doi:10.3390/ijms22052684_

Round 1
Reviewer 1 Report
This is an appropriate review summarizing medical details related to mediator symptoms and anaphylaxis in children with mastocytosis. Mastocytosis is a rare disease of mast cell accumulation in most cases in the skin and which resolves sponataneously in most cases during puperty (60 to 80%). The authors describe in detail the mediators and the related symptoms involved in causing the symptoms making the case of the most severe and sometimes life-threatining form, which is anaphylaxis. The finalize the review by discussing diagnosis and treatment options.
The review is well written and publishable.
The only major comment I have is that related to the presented two tables the citation place is not always appropriaye. Thus on line 153, they indicate that anaphylaxis range between 0 and 9%. However Table 1 does not show this. Similarly they cite table 2 for showing an increased risk for hymenoptera stings. However, Table 2 does nor show this.
Concerning hereditary alpha-tryptasemia could the authors speculate why increased tryptase represents a risk factor for anaphylaxis?
Likewise, how do the authors explain treatment efficacy of omalizumab. Is this associated generally with an allergy or is it independent?
Author Response
This is an appropriate review summarizing medical details related to mediator symptoms and anaphylaxis in children with mastocytosis. Mastocytosis is a rare disease of mast cell accumulation in most cases in the skin and which resolves sponataneously in most cases during puperty (60 to 80%). The authors describe in detail the mediators and the related symptoms involved in causing the symptoms making the case of the most severe and sometimes life-threatining form, which is anaphylaxis. The finalize the review by discussing diagnosis and treatment options.
The review is well written and publishable.
RE: We thank this reviewer for the favourable comments
The only major comment I have is that related to the presented two tables the citation place is not always appropriate. Thus on line 153, they indicate that anaphylaxis range between 0 and 9%. However Table 1 does not show this. Similarly they cite table 2 for showing an increased risk for hymenoptera stings. However, Table 2 does not show this.
RE: Thank you for bringing this up. We agree with the reviewer that the citation places of the tables were not appropriate. This was a mistake. We have therefore re-ordered the tables in our revised manuscript. First, on line 158 (formerly 153) we have now put “Table 1: Frequency of anaphylaxis in children with mastocytosis in the literature” and the Table 1 backs up the statement on the frequency of anaphylaxis now. Table 2 “Risk factors for anaphylaxis in children with mastocytosis” has been correctly positioned to line 235 now after the sentence “However, important risk factors have been identified for anaphylaxis in children (Table 2)”. The increased risk for anaphylaxis to hymenoptera stings is now discussed in lines 171 ff. with Table 1 and literature backing up this statement (“The most prominent trigger of anaphylaxis in adults with mastocytosis is insect venom (43,47,48), whereas surprisingly in children with mastocytosis, hymenoptera stings are not a frequent trigger and there is no evidence for an increased risk in this patient group (Table 1) (34,49)”).
Concerning hereditary alpha-tryptasemia could the authors speculate why increased tryptase represents a risk factor for anaphylaxis?
RE: We agree that the paper would improve from a discussion about the potential mechanisms through which hereditary alpha-tryptasemia could contribute to an increased risk for anaphylaxis. Following the suggestion of the reviewer, we have included such a discussion: in fact, we speculate that an increased amount of tryptase may also lead to an increased load of other mediators (per mast cell) that are rapidly released by these cells through an anaphylactic event. An alternative explanation, recently been proposed by Le et al (JEM 2019;216:2348-2361), is that hetero-tetramers of tryptase, composed of 2α- and 2β-tryptase protomers (α/β-tryptase), activate protease-activated receptor-2 on smooth muscle, neurons, and endothelium and increase mast cell susceptibility to vibration-triggered degranulation by cleaving the α subunit of the EGF-like module-containing EMR2. We have included these speculations in our revised manuscript.
Likewise, how do the authors explain treatment efficacy of omalizumab. Is this associated generally with an allergy or is it independent?
RE: We agree that it would also be of interest to know whether efficacy of omalizumab in mastocytosis and MCAS is related to an allergic disorder or at least specific IgE. Omalizumab efficacy is not only restricted to allergic responses, but also abrogates non-allergic chronic spontaneous urticaria. It is believed that this effect is related to downregulation of FcεRI receptors on basophils and mast cells, however, its detailed mechanism of action is not currently understood. On the other hand, most anaphylactic reactions in patients with mastocytosis and/or MCAS can now be explained by (often occult forms of an) IgE-dependent allergy. We have included these considerations in our revised manuscript.
Reviewer 2 Report
Brockow et al present here a comprehensive review on clinical symptoms due to mast cell mediator release in paediatric patients with mastocytosis. The review is concise and well-structured, and I have no major criticisms.
However, I think the following paper should be added to reference 10:
- The 2016 revision to the World Health Organization classification of myeloid neoplasms and acute leukemia - Daniel A Arber et al – Blood – 2016 (PMID: 27069254).
Author Response
Brockow et al present here a comprehensive review on clinical symptoms due to mast cell mediator release in paediatric patients with mastocytosis. The review is concise and well-structured, and I have no major criticisms.
However, I think the following paper should be added to reference 10:
- The 2016 revision to the World Health Organization classification of myeloid neoplasms and acute leukemia - Daniel A Arber et al – Blood – 2016 (PMID: 27069254).
RE: Following the suggestion of the reviewer we have included this reference in our revised manuscript.
Reviewer 3 Report
This is a complete review of the clinical symptoms of mast cell acTivation diseases in children, which covers all aspects related to the diagnosis, follow-up and therapy.
Despite the disease is considered a rare disease due to its low prevalence, mediator-related symptoms can affect the pediatric population, so it feels necessary that medical doctor from different disciplines should have a basic knowledge of the diagnosis and management.
This review is a very comprehensive work that can help to clarify the sometimes difficult approach to the diagnosis and follow-up of a rare and complex disease as mastocytosis.
In my opinion, this is a suitable journal to be published so many doctors will have the opportunity to read it.
Author Response
RE: We thank this reviewer for the favourable comments